# Streamlining Temporal Formal Verification over Columnar Databases

**Giacomo Bergami** 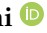

School of Computing, Faculty of Science, Agriculture and Engineering, Newcastle University, Newcastle upon Tyne NE4 5TG, UK; giacomo.bergami@newcastle.ac.uk

**Abstract:** Recent findings demonstrate how database technology enhances the computation of formal verification tasks expressible in linear time logic for finite traces ($\text{LTL}_f$). Human-readable declarative languages also help the common practitioner to express temporal constraints in a straightforward and accessible language. Notwithstanding the former, this technology is in its infancy, and therefore, few optimization algorithms are known for dealing with massive amounts of information audited from real systems. We, therefore, present four novel algorithms subsuming entire $\text{LTL}_f$ expressions while outperforming previous state-of-the-art implementations on top of KnoBAB, thus postulating the need for the corresponding, leading to the formulation of novel $\texttt{xtLTL}_f$-derived algebraic operators.

**Keywords:** temporal formal verification; columnar databases; verified artificial intelligence; linear time logic for finite traces

## 1. Introduction

Grounded in formal methods, verified artificial intelligence [1] is concerned with defining, designing, and verifying systems represented mathematically. In context-free data, this focuses on a system $\mathfrak{S}$ to be verified through properties described in $\Phi$, while the model of the environment $\mathfrak{E}$ is neglected. In this regard, a *formal verification* task ascertains whether a given system complies with a specification $\mathfrak{S} \models \Phi$. In the context of business process management, we can consider *model* [2], *conformance* [3], or *compliance* [4,5] *checking* as all synonyms of the former. Concerning temporal data, we focus our attention on systems described as logs, a collection of temporally ordered records (i.e., *traces*) of observed and completed (or aborted) labelled activities unravelling one possible run of a process. These real-world processes might include the auditing of malware in terms of system calls being invoked [6,7], records describing patients' hospitalization procedures [8–10], as well as transactions between producers and retailers through a brokerage system [11]. As an example, each trace of a log can describe three distinct patient registration events at an emergency department (ED) [12] as given by the following log expressed in terms of the activity labels associated to our events:

$$
\begin{aligned}
\mathfrak{S} = \{ & \langle \texttt{registration}, \texttt{examination}, \texttt{discharge} \rangle, \\
& \langle \texttt{registration}, \texttt{redirection}, \texttt{clinical test}, \\
& \quad \texttt{examination}, \texttt{discharge} \rangle, \\
& \langle \texttt{registration}, \texttt{redirection}, \texttt{examination}, \\
& \quad \texttt{discharge} \rangle \}
\end{aligned}
\tag{1}
$$

In all these contexts, a formal verification task returns whether the current instances of the processes being collected as traces of a log $\mathfrak{S}$ abide by specific temporal quality requirements $\Phi$ while determining which temporal constraints in $\Phi$ are explicitly violated. Linear Temporal Logic over Finite traces ($\text{LTL}_f$, Section 2.1) [13] can be used to express these temporal specifications $\Phi$. This logic is defined as linear since it assumes there is only

one future possible event immediately following a given event in a sequence of events of interest. Such low-level semantics are then exploited to give the semantics of temporal templates, expressing occurring temporal correlations of interest; the present paper will discuss Declare [14].

The emerging area of temporal big data analytics, having data with time as a first-class citizen, makes the need to efficiently process the aforementioned tasks more pressing [15,16]. In such real scenarios, adopting relational databases provides an ideal setting for dealing with such temporal data [17]. This also includes the storage and querying of numerical time series [18], or considering different versions in time of entities and relationships represented in the relational model [19–22]. In recent years, researchers have demonstrated that time series can be represented as traces via time series segmentation by discretizing the variation in time series into discrete, observable, linear events that are distinct from each other, enabling identification of a system's transitional states [23] as well as variations in the values associated with time series [24]. As a result of such segmentation, pattern searches can now be run using streamlined approaches. $LTL_f$ has now been applied to a widespread set of applications in real use case scenario contexts, such as controlling actuation upon sensing the environment in Industry 4.0 settings [25] as well as for the verification of smart contracts [26], for which this technology proved to be effective for verified artificial intelligence. The large adoption of such formal language pushes us to focus on this well-known and consolidated language [13,27].

In the context of formal specification tasks expressed in $LTL_f$, recent research clearly remarked on the inadequacy of off-the-shelf row-based relational databases and SQL as a query language for expressing $LTL_f$ temporal constraints, as it clearly showed that a customized relational algebra for expressing formal specification (eXTended $LTL_f$, $xtLTL_f$ [28]) and query plan minimizing the running of sub-queries [29] running on customized column-based storage (KnoBAB [28,30]) outperformed the previous solution. The main benefit of this approach is that any $LTL_f$ can be directly expressed in terms of $xtLTL_f$, while high-level and human-readable temporal constraints expressed through temporal clauses can be directly specified in a semantics query at warm-up, thus allowing the support of any declarative temporal language (`queryplan` in Figure 1). As this line of research is in its infancy, very few algorithms for efficiently running $xtLTL_f$ are known. We now remark on two use cases addressed for the first time in the present paper.

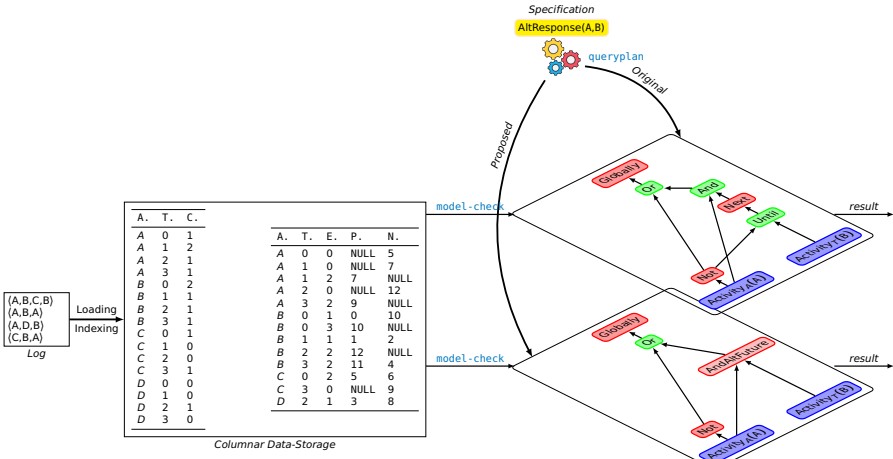

**Figure 1.** High-level representation of the KnoBAB query plan for running a AltResponse(`A`,`B`) for different specifications over a pre-loaded log within a columnar data-storage. After loading and indexing some traces stored as a log, we obtain a columnar data storage. At warm-up time, we can specify a `queryplan` which, at formal verification (`model-check`) time, converts a Declare specification into a $xtLTL_f$ query plan. As KnoBAB supports multiple `queryplan`s at once, we can run the same formal verification task over different resulting query plans.

First, due to their formulation, some of the logical operators such as the timed until operator $\text{UNTIL}^{\tau}_{\textbf{True}}(\varphi, \varphi')$ ($\varphi \mathcal{U} \varphi'$ in $\text{LTL}_f$) are associated with very high computational complexity, as it prescribes that the occurrence of at least one future event matching a $\varphi'$ condition per trace shall always be preceded by events matching $\varphi$. Under the occasions that this temporal post-condition shall be considered only after determining the occurrence of a first event $\varphi''$, this could drastically reduce the amount of computation associated with the overall task. This is not taken into account in our previous implementation in KnoBAB, as it computed a union between the cases where $\varphi''$ does not occur and the ones where $\varphi''$ occurs, for which the evaluation of $\text{UNTIL}^{\tau}_{\textbf{True}}(\varphi, \varphi')$ is extended to any event occurring of the trace. Walking in the footsteps of relational algebra, where $\theta$-joins are expressed as the combination of natural joins [31] or cross-products [32] with $\theta$-selections and join operations can be streamlined through cogrouping [33], we then propose similarly derived operators, combining the matching of a given pre-condition with the subsequent requirement that all the intermediate events should meet the alternance requirements dictated by $\text{UNTIL}^{\tau}_{\Theta}$. This paper will then contextualize the need for such derived operators for two specific Declare temporal templates, AltPrecedence and AltResponse, thus substantiating the interest in these temporal patterns from the current literature (Table 1).

**Table 1.** Declare templates as exemplifying clauses. *A* (*B*) represents the *activation* (*target*) condition as an activity label.

| | Exemplifying Clause ($c_l$) | Natural Language Specification for Traces | $\text{LTL}_f$ Semantics ($[\![c_l]\!]$) |
|---|---|---|---|
| *In this paper* | ChainPrecedence $(A, B)$ | The activation is immediately preceded by the target. | $\square(\bigcirc A \Rightarrow B)$ |
| | ChainResponse $(A, B)$ | The activation is immediately followed by the target. | $\square(A \Rightarrow \bigcirc B)$ |
| | AltResponse $(A, B)$ | If activation occurs, no other activations must happen until the target occurs. | $\square(A \Rightarrow \bigcirc(\neg A \, \mathcal{U} \, B))$ |
| | AltPrecedence $(A, B)$ | Every activation must be preceded by a target without any other activation in between | $\neg B \, \mathcal{W} \, A \wedge \square(A \Rightarrow \bigcirc(\neg A \, \mathcal{W} \, B))$ |
| *Not subject to optimization in this paper* | Init $(A)$ | The trace should start with an activation | $A$ |
| | Exists $(A, n)$ | Activations should occur at least *n* times | $\Diamond(A \wedge \bigcirc([\![\text{Exists}(A, n-1)]\!])_{n>0})$ |
| | Absence $(A, n+1)$ | Activations should occur at most *n* times | $\neg[\![\text{Exists}\,(A, n+1)]\!]$ |
| | Precedence $(A, B)$ | Events preceding the activations should not satisfy the target | $\neg B \, \mathcal{W} \, A$ |
| | Choice $(A, A')$ | One of the two activation conditions must appear. | $\Diamond A \vee \Diamond A'$ |
| | Response $(A, B)$ | The activation is either followed by or simultaneous to the target. | $\square(A \Rightarrow \Diamond B)$ |
| | RespExistence $(A, B)$ | The activation requires the existence of the target. | $\Diamond A \Rightarrow \Diamond B$ |
| | ExlChoice $(A, A')$ | Only one activation condition must happen. | $[\![\text{Choice}(\texttt{A}, \texttt{A'})]\!] \wedge [\![\text{NotCoExistence}(\texttt{A}, \texttt{A'})]\!]$ |
| | CoExistence $(A, B)$ | RespExistence, and vice versa. | $[\![\text{RespExistence}(\texttt{A}, \texttt{B})]\!] \wedge [\![\text{RespExistence}(\texttt{B}, \texttt{A})]\!]$ |
| | Succession $(A, B)$ | The target should only follow the activation. | $[\![\text{Precedence}(\texttt{A}, \texttt{B})]\!] \wedge [\![\text{Response}(\texttt{A}, \texttt{B})]\!]$ |
| | ChainSuccession $(A, B)$ | Activation immediately follows the target, and the target immediately preceeds the activation. | $\square(A \Leftrightarrow \bigcirc B)$ |
| | NotCoExistence $(A, B)$ | The activation nand the target happen. | $\neg(\Diamond A \wedge \Diamond B)$ |
| | NotSuccession $(A, B)$ | The activation requires that no target condition should follow. | $\square(A \Rightarrow \neg \Diamond B)$ |

*Legend*: Globally: $\square\phi$, Next: $\bigcirc\phi$, Implication: $\phi \Rightarrow \phi'$, Until: $\phi' \, \mathcal{U} \, \phi$, Weak Until: $\phi \, \mathcal{W} \, \phi'$, Future: $\Diamond\phi$.

**Example 1.** *AltResponse*($\texttt{A}, \texttt{B}$) *requires that, when* $\texttt{A}$ *occurs,* $\texttt{B}$ *shall occur anytime in the future while no other* $\texttt{A}$ *shall occur in between. In* $\texttt{xtLTL}_f$*, this can be expressed as* $\square(\neg\texttt{A} \vee (\texttt{A} \wedge \bigcirc(\neg\texttt{A} \, \mathcal{U} \, \texttt{B})))$ *(Original in Figure 1). On the other hand, the present paper shows that, by replacing* $\texttt{A} \wedge \bigcirc(\neg\texttt{A} \, \mathcal{U} \, \texttt{B})$ *with a single operator, we obtain a significant reduction in running time by reducing the amount of result scans and data allocations. This is possible by providing a different (Proposed)* $\texttt{xtLTL}_f$ *queryplan while implementing AndAltFuture as a novel operator. This difference is remarked in the two resulting query plans in Figure 1.*

Second, temporal constraints requiring that events abiding by a $\varphi$ specification shall always precede (or follow) other events abiding by $\varphi'$ are currently implemented in KnoBAB by equi-joining all the events matching $\varphi$ with the ones matching $\varphi'$, while the predicate is $i = i' \wedge j = j - 1$ (or $i = i' \wedge j = j' + 1$), where $i$ (or $i'$) and $j$ (or $j'$) are, respectively, referring

to the trace id and event id associated to a record coming from the first (or second) operand (see $\text{And}_\Theta^\tau$ xtLTL$_f$ in Section 2.2.2). Even this implementation can be further boosted by minimizing the data table access to just one operator (e.g., $\varphi$) for directly accessing the immediately preceding or following events within the relational database and checking whether they abide by $\varphi'$. Even this second observation is motivated by the existence of ChainResponse and ChainPrecedence Declare templates, thus requiring the definition of novel derived operators for performance purposes.

To support our research claims, we extend (https://github.com/datagram-db/knobab/releases/tag/v2.3, accessed on 3 January 2024) the current implementation of KnoBAB [34], a column-oriented main memory DBMS supporting formal verification and specification mining tasks by defining relational operations for temporal logic and customary mining algorithms. Despite this being a main memory engine, it currently supports intra-query parallelism and hybrid algorithms (Section 2.2.1). To our knowledge, no other database management system for temporal formal verification over LTL$_f$ provides these features, for which we choose to extend such a system. Furthermore, KnoBAB already proved to consistently outperform previous state-of-the-art algorithms on both tasks [35], thus including competing approaches interpreting the same temporal constraints over SQL and row-oriented relational database architecture [17]. After providing a brief literature overview on the landscape of formal verification for temporal data (Section 2), we outline the following main contributions leading to the our performance analysis result for our newly proposed xtLTL$_f$ operators:

- We formally introduce the novel temporal operators optimizing the aforementioned scenarios in the context of Declare as a declarative language for formal verification (Section 3).
- We describe the implementation of the aforementioned operators over the KnoBAB architecture leveraging columnar-oriented main memory storage (Section 4).
- We present experimental results to evaluate the effectiveness of such newly introduced operators in the context of formal verification in Declare (Section 5).

## 2. Related Works

### 2.1. Languages for Temporal Formal Specifications

#### 2.1.1. LTL$_f$

Taking the possible worlds as finite traces, LTL$_f$ is a well-established extension of modal logic with modalities referring to time; it assumes that all the events of interest are fully observable and therefore deterministic and that, for each occurring event, they should be immediately followed by at most one event. This entails that the $i$-th trace $\sigma^i$ in a log $\mathfrak{S}$ can be considered as a sequence of $n$ totally ordered events $\sigma_0^i \ldots \sigma_{n-1}^i$, where each event $\sigma_j^i$ is associated to a single activity label $\lambda(\sigma_j^i) \in \Sigma$ [3]. When events are associated to a payload represented as a key-value association $\varsigma(\sigma_j^i)$, we refer to such logs as *dataful* and as *dataless* otherwise. In the eventuality of the former, such payloads can be represented as finite functions $V^K$, where $K$ is the set of the keys and $V$ is the overall set of non-NULL values. Concerning our datasets of interest, we only consider ones where trace events are not associated with a data payload, and therefore even such logs can be considered as *dataless*. On the other hand, with reference to Equation (1), event payloads can store patient information, thus registering the recorded medical condition being observed [28]; in the context of good brokerage, such payload might contain the relevant contract information between the supplier and the customer which are required to be respected (e.g., delivery times), as well as the location of the goods, their number, and quality [11].

LTL$_f$ semantics is usually defined in terms of First-Order Logic [36]; more informally, Next ($\bigcirc\phi$) requires $\phi$ to occur from the subsequent temporal step, Globally ($\square\phi$) that $\phi$ always holds from the current instant of time, Future ($\lozenge\phi$) that $\phi$ must eventually hold, and Until $\phi\,\mathcal{U}\,\phi'$ that $\phi$ must hold until the first occurrence of $\phi'$ does. Weak Until is a *derived operator* for $\varphi\mathcal{W}\varphi' := \varphi\mathcal{U}\varphi' \vee \square\varphi$, while the logical implication can be rewritten

as $\varphi \Rightarrow \varphi' := (\neg\varphi) \lor (\varphi \land \varphi')$. Please observe that LTL$_f$ does not provide full support for handing data correlation conditions between operands of binary operations, as it only supports the declaration of data conditions that can be applied to one single event [3]. To the best of our knowledge, xtLTL$_f$ (Section 2.2.2) is the only extension of this language supporting data payload correlation across events matched by both arguments of the binary operator, thus providing a complete *dataful* support.

### 2.1.2. Declare

Declare [14,37] provides a human-readable declarative language on top of LTL$_f$ (first column of Table 1), where each template is associated with a specific LTLf formula (third column), which can be instantiated with arbitrary activity labels. We refer to the instantiation of such templates via activity labels in a finite set $\Sigma$ as *(declarative) clauses*. Declare circumscribes the set of all the possible behaviors expressible in LTL$_f$ to the ones of interest over a set of possible $\Sigma$; Table 1 recalls some of the most used templates while remarking on the four templates of interest optimized in the present paper.

At the time of writing, Declare expresses specifications $\Phi$ as a set of clauses $c_l$ being usually associated with an LTL$_f$ semantics $[\![c_l]\!]$; in this context, a trace $\sigma \in \mathfrak{S}$ satisfies a Declare specification $\Phi$ if it jointly satisfies all the clauses associated to the specification. If these clauses can be characterized by a precondition which, if satisfied by some event, imposes the occurrence of a post-condition, then we refer to these as *activation* and *target* conditions, respectively. Please observe that post-conditions are considered as such merely in terms of causal implication (i.e., $\Rightarrow$) and not necessarily in temporal terms, e.g., while ChainResponse requires the target to immediately follow any existing activation, ChainPrecedence requires that the targeted event shall instead precede the activation. Please consider that Declare clauses do not necessarily reflect association rules, as the latter do not provide temporal constraints correlating the activation of activation and target conditions. In this paper, we focus on Declare clauses only predicating over the events' activity labels, which are then referred to as *dataless*; on the other hand, *dataful* Declare clauses can express data payload conditions over both activation and target conditions, as well as representing $\Theta$ payload correlation conditions between activating and targeted conditions [28]. Thus, both clauses and logs are referred to *dataful* otherwise.

Despite the fact that the four clauses of interest in Table 1 might appear to express similar behavior, they express substantially different concepts. Table 2 provides four traces distinguishing the behavior of such four templates, the validity of which can be easily controlled by transforming the associated LTL$_f$ formulæ into a DFA (http://ltlf2dfa.diag.uniroma1.it/dfa, accessed on 3 January 2024).

**Table 2.** Traces from the Log in Figure 1 distinguishing the temporal behavior of the Declare clauses of interest in this paper, where each trace $\sigma_0^i \ldots \sigma_{n-1}^i$ is expressed in terms of their associated activity labels, $\langle \lambda(\sigma_0^i), \ldots, \lambda(\sigma_{n-1}^i) \rangle$. ✓(and ✗) remarks a trace satisfying (violating) a corresponding clause.

| Traces | ChainResponse(A,B) | ChainPrecedence(B,A) | AltResponse(A,B) | AltPrecedence(B,A) |
|---|---|---|---|---|
| $\langle$A,B,C,B$\rangle$ | ✓ | ✗ | ✓ | ✗ |
| $\langle$A,B,A$\rangle$ | ✗ | ✓ | ✗ | ✗ |
| $\langle$A,D,B$\rangle$ | ✗ | ✗ | ✓ | ✗ |
| $\langle$C,B,A$\rangle$ | ✗ | ✗ | ✗ | ✓ |

### 2.2. KnoBAB and xtLTL$_f$

*We now summarize our previous contributions on temporal formal verification tasks run over our proposed main memory columnar database, KnoBAB.*

### 2.2.1. KnoBAB

*KnoBAB* [28,34] is a column database store tailored for both loading *dataful* logs being represented in XES [38] and *dataless* ones described as a tab-separated file. This outper-

formed the previous state of the art in terms of both specification mining [39] and formal verification [35] tasks on tailored non-database solutions.

Logical and Physical Model

The resulting column-based relational database is then represented through some tables having fixed schema independently from its data representation. As the present paper focuses on dataless datasets, we describe in this paper just two of those; Table 3 describes the relational representation of the log presented in Equation (1). The ActivityTable (Table 3a) lists each trace event of a given log, where records are sorted in ascending order for activity label, trace id, and event id. Cells under the `Prev` (and `Next`) column store a pointer to the record representing the immediately preceding (and following) event in the same trace if any. After mapping each existing activity label in the log a to a unique natural number $\beta(a)$, we can define a primary dense and clustered index that can be accessed in $O(1)$ time as it is an array of offset pointers. We also define a secondary index structured as a block of two records, associating each trace in the log to the first and last trace event; given that all the traces are associated with a unique natural number, this index can also be accessed on $O(1)$ time by trace id. The CountTable (Table 3b), also created at loading time like the previous, merely lists the number of occurrences of each activity label per trace and can be used to determine the absence or presence of an event with a given activity label per trace.

Formal Verification Tasks over Query Plans

In spite of the ActivityTable also appearing in SQLMiner's log representation [17] (except for the `Prev` and `Next` columns), this still used an off-the-shelf relational database engine and a translation of Declare specification into SQL for carrying out formal verification tasks over a dataless log. KnoBAB showed a new pathway for enhancing temporal queries over customary main memory relational database through the combined provision of both ad hoc relational operators expressing LTL$_{\mathrm{f}}$ over relational tables (xtLTL$_{\mathrm{f}}$) and the definition of a query plan represented as a rooted DAG where shared subqueries are computed only once [29]. This was sensibly different from competing approaches [40,41] also relying on main memory engines where, instead, the query plan associated to a formal verification task is always expressed in terms of trees, thus not allowing the detection of shared sub-expressions to be merged to avoid wasteful recomputations. As vertices for a DAG can be sorted topologically, we can obtain for free the scheduling order in which the operators must be executed and, by associating each node a maximum distance value from the root, we can safely run in parallel all the operators laying at the same depth level, as all the previously called operators will pertain their information in an intermediate cache, thus achieving intraquery parallelism as a free meal [28]. This parallelization approach greatly differs from straightforward parallelization algorithms known in the Business Process Management area, where they simply run each declarative clause occurring in the specification in a separate thread [35]. In addition to the former, KnoBAB guarantees efficient access to the tables through the provision of specific indexing data structures such as primary indices for directly accessing the blocks of the table concerning a specific activity label as well as the provision of secondary indices mapping a specific trace id $i$ and event id $j$ for $\sigma_j^i$ into a table offset. KnoBAB outperformed SQLMiner run over PostgreSQL within two to five orders of magnitude, thus demonstrating the inadequacy of using customary relational operators for computing temporal tasks over relational databases.

**Table 3.** KnoBAB representation for the dataless log in Equation (1). (**a**) ActivityTable; (**b**) CountTable.

(**a**)

| ActivityLabel | TraceId | EventId | Prev | Next |
|---|---|---|---|---|
| Clinical Test | 1 | 2 | 7 | 5 |
| Discharge | 0 | 2 | 4 | NULL |
| Discharge | 1 | 4 | 5 | NULL |
| Discharge | 2 | 3 | 6 | NULL |
| Examination | 0 | 1 | 9 | 1 |
| Examination | 1 | 3 | 0 | 2 |
| Examination | 2 | 2 | 8 | 3 |
| Redirection | 1 | 1 | 10 | 0 |
| Redirection | 2 | 1 | 11 | 6 |
| Registration | 0 | 0 | NULL | 4 |
| Registration | 1 | 0 | NULL | 7 |
| Registration | 2 | 0 | NULL | 8 |

(**b**)

| ActivityLabel | TraceId | Count |
|---|---|---|
| Clinical Test | 0 | 0 |
| Clinical Test | 1 | 1 |
| Clinical Test | 1 | 0 |
| Discharge | 0 | 1 |
| Discharge | 1 | 1 |
| Discharge | 2 | 1 |
| Examination | 0 | 1 |
| Examination | 1 | 1 |
| Examination | 2 | 1 |
| Redirection | 0 | 0 |
| Redirection | 1 | 1 |
| Redirection | 2 | 1 |
| Registration | 0 | 1 |
| Registration | 1 | 1 |
| Registration | 2 | 1 |

KnoBAB enables the specification of user-defined template names in terms of $xtLTL_f$ operators through a `queryplan ''semanticsname'' {...}` query, thus allowing the co-presence of multiple possible definitions of declarative clauses. Then, we can select the most appropriate semantics while carrying out the formal verification task by specifying such a name, e.g., `model-check ...plan ''semanticsname''...` This then enables us in this paper to test multiple possible specifications of Declare clauses without necessarily recompiling the database's source code.

Walking in the footsteps of the BAT algebra for columnar databases [42], each of the novel temporal operands for $xtLTL_f$ (Section 2.2.2) not requiring direct data access to the aforementioned KnoBAB tables both accepts as an input and returns a uniform data representation $\rho$ with schema:

$$\text{IntermediateRepresentation}(\text{TraceId}, \text{EventId}, \text{Witnesses}(\text{Tag})) \qquad (2)$$

where the first (and second) argument refers to the trace (and event) id matching a specific temporal condition of choice, while `witnesses` represents the relevant activated or targeted conditions occurring from the position `EventId` in a given `TraceId` trace onwards via a tagged extension of semiring provenance [43]; such tags mainly refer to the distinction between activated and targeted events, respectively $A$ and $T$. Dataful matching occurring between witnessed activated $A(i)$ and targeted events $T(j)$ certified via a $\Theta$ binary predicate

are represented as $M(i, j)$. Matches can be represented as semiring products, while the listing of all the activated, targeted, and matched events can be represented as a semiring sum; the latter is simply rendered as a list. As the table is sorted by trace id and event id by design for any given activity label, such intermediate representation also returns trace entities sorted by ascending trace id and event id.

### 2.2.2. xtLTL$_f$

*We now discuss some xtLTL$_f$ operators of relevance for the current paper. By using KnoBAB as a computational model, we can also discuss the time complexity associated with such operators.* While LTL$_f$ operators can mainly be used to establish a yes/no question about whether a single trace abides by some temporal specification, an xtLTL$_f$ expression returns all the traces in the log conforming to a temporal specification by composing the trace events as records through temporal operations. Furthermore, the latter can also be directly exploited to express confidence, maximum satisfiability, and support metrics similar to association rules. So to better support future explainable temporal AI tasks, xtLTL$_f$ also carries out information concerning activated/targeted events justifying the algorithmics' outcome, while the cache associated to the leaves can be analyzed so as to check which events were activated/targeted without necessarily satisfying the temporal requirements computed through xtLTL$_f$.

Table Access ("Leaf") Operators

We determine all the events being associated with a specific activity label through the ActivityLabel's primary block index and express the outcome of this retrieval in terms of intermediate representation:

$$\text{Activity}_{A/T}^{\mathfrak{S},\tau}(\mathsf{a}) = \{\langle i, j, \{A/T(j)\}\rangle \,|\exists \pi, \phi. \,\langle \mathsf{a}, i, j, \pi, \phi\rangle \in \text{ActivityTable}\}$$

where $A/T$ provides the optional tags for remarking the matching event of interest as being part of an activation/target condition. By associating each activity label a with a unique natural number $\beta(\mathsf{a})$, we can now seek the presence of events with label a in $O(1)$ time and retrieve all the events $\#\mathsf{a} \ll |\mathfrak{S}|$ associated to such a label. If, on the other hand, we are interested in events matching a specific data predicate $q$, we define the following operator:

$$\text{Atom}_{A/T}^{\mathfrak{S},\tau}(\mathsf{B}, q) = \{\langle i, j, A/T(j)\rangle \mid q(\sigma_j^i) \wedge \lambda(\sigma_j^i) = \mathsf{B}\}$$

Despite the fact that this might appear as a simple selection operation, the atomization of a predicate into mutually exclusive data conditions required for both minimizing the data access to the tables holding the key-value payload associations within the dataful events and merging multiple equivalent sub-expressions into one makes both its associated query plan and its actual formal definition quite convoluted. As describing this is not the major purpose of the paper, we refer to [28] for any further information. By accessing the secondary index of the ActivityTable, we can collect the last events for each trace in linear time over the log's size $O(|\mathfrak{S}|)$ using the following operator:

$$\text{Last}_A^{\mathfrak{S},\tau} = \{\langle i, |\sigma^i|, \{A(|\sigma^i|)\}\rangle \,|\exists \mathsf{a}, \pi. \,\langle \beta(\mathsf{a}), i, |\sigma^i|, \pi, \texttt{NULL}\rangle \in \text{ActivityTable}\}$$

Unary Operators

We discuss the main difference between operators' execution in xtLTL$_f$ from corresponding ones in LTL$_f$; the latter computes semantics from the first occurring operator appearing in the formula towards the leaves, whereas the former assumes intermediate results coming from the leaves. In this, the downstream operator is completely agnostic about the semantics associated with the upstream operator, so it must combine the intermediate results appropriately. Therefore, the $\text{Next}(\rho)$ (timed) xtLTL$_f$ unary operator returns all the events $\sigma_j^i$ witnessing the satisfaction of an activation, target, or correlation condition being returned by a downstream operator as an intermediate result $\rho$, while $\bigcirc\varphi$ will simply

increment the internal time counter over $\varphi$, thus determining the time from which to assess the specification in $\varphi$.

Due to this structural discrepancy in the order of computation, $\mathtt{xtLTL_f}$ must distinguish *timed* operators (assessing the occurrence of a specification sub-expression anytime in the trace) from the *untimed* operators (determining the properties holding from the beginning of the trace). The aforementioned $\mathtt{xtLTL_f}$ operator can therefore be expressed as follows:

$$\mathsf{Next}^\tau(\rho) = \{\ \langle i, j-1, L\rangle \mid \langle i, j, L\rangle \in \rho, j > 0\ \}$$

This operator can then be computed in linear time over the size of the input, i.e., $O(|\rho|)$. On the other hand, the timed negation operator $\mathsf{Not}^\tau(\rho)$ subtracts from the universal relation, being all the events occurring in any trace, the ActivityTables events appearing in $\rho$ while still guaranteeing the return of the records in ascending order for trace and event id. Given $\epsilon$, the maximum trace length, this operator takes at most $O(|\mathfrak{S}|\epsilon)$ time by assuming $|\rho| \ll |\mathfrak{S}|\epsilon$. The globally timed operator prescribes to return a $\langle i, j, L\rangle \in \rho$ if also all the subsequent events within the same trace are in $\rho$, and can be computed in $O(|\rho| \log |\rho|)$ time by starting scanning the events from the last occurring in the trace.

Binary Operators

We now stress further differences between $\mathtt{xtLTL_f}$ and $\mathtt{LTL_f}$ in terms of binary operators. While $\mathtt{xtLTL_f}$ can express dataful matching conditions between activation and target conditions, $\mathtt{LTL_f}$ can only express properties associated with one single event at a time through atoms. In these regards, timed logical conjunction ($\mathsf{And}^\tau_\Theta(\rho, \rho')$) extending the logical conjunction in $\mathtt{LTL_f}$ with a binary match condition $\Theta$ over the event's payloads can be expressed as a nested $\Theta$-join returning the records from both operands having the same trace id and event id, while all the pairs of witnessed events satisfying an activation $A(i)$ and target $T(j)$ conditions from the matching record shall satisfy the $\Theta$ matching condition when provided; the matching is then registered with $M(i, j)$. Timed logical disjunction ($\mathsf{Or}^\tau_\Theta(\rho, \rho')$) can be similarly expressed through a full outer $\Theta$-join. Given that the ActivityTable is pre-sorted at indexing time, we can efficiently implement such algorithms through sorted joins. As these can be computed with a joint linear scan of both operands, both operators have at most a time complexity in $O(|\rho| + |\rho'|)$. The timed until operator ($\mathsf{Until}^\tau_{\mathbf{True}}(\rho, \rho')$) for $\Theta = \mathbf{True}$ is defined similarly to the corresponding $\mathtt{LTL_f}$ operator; it returns all the events within a given log trace in the second operand and the events from the first operand if all the immediately following events until the first occurrence of an event in the second operand also belong to the first:

$$\mathsf{Until}^\tau_{\mathbf{True}}(\rho, \rho') = \rho' \cup \{\ \langle i, k, L \cup L'\rangle \mid \exists j > k.\ \langle i, j, L\rangle \in \rho', (\forall k \leq h < j.\ \langle i, h, L'\rangle \in \rho)\ \}$$

This can be computed in $O(|\rho|^2 |\rho'|)$ time in its worst-case scenario. The in-depth discussion concerning the formal definition of such an operator when matching a non-trivially true matching condition $\Theta$ is deferred due to its technicalities and can be retrieved from the original paper [28].

### 2.3. Algebraic Specification for Queries

We now compare $\mathtt{xtLTL_f}$ with other long-standing definitions of temporal operators regarding database temporal representations.

Current research [17] outlined the possibility of loading logs composed of multiple traces within row-based relational databases while providing a direct translation of *dataless* Declare-driven formal verification and specification mining tasks into SQL [44]. Our previous research remarked on the inefficiency of directly expressing temporal formal verification tasks on top of off-the-shelf relational databases, thus motivating the definition of a novel query plan specification directly exploiting temporal algebra operators, $\mathtt{xtLTL_f}$ [28]. As SQL queries are translated into query plans where each operator expresses an implementation of a relational algebra operator, this demonstrates the overall inefficiency of

exploiting traditional relational algebra for representing temporal queries. Please observe that $LTL_f$ temporal requirements cannot be expressed in traditional relational algebra without aggregation operators while not naturally assuming a columnar database storage. Therefore, traditional relational algebra cannot be directly exploited to predicate about the necessity or the eventuality of a given event to occur without any further extension.

For all these considerations, our proposed algebra more resembles BAT from MonetDB [42,45], where the intermediate result output for each operator records the table's record being selected, without necessarily carrying out values stored within the specific row. Given the specificity of our scenario, our intermediate results carry the trace id and the event id as unique record identifiers. We further had to extend this representation to possibly carry out the activated and targeted events as witnesses of the computation's correctness, providing explainable justifications for the computation, and correctly expressing $\Theta$ predicates over dataful logs. $xtLTL_f$ then provides a required extension of such a representation for new computation needs.

Concerning Allen's algebra for temporal intervals [46], we can *first* see that such algebra considers events as temporal intervals that might also be overlapping, while $xtLTL_f$ inherits the same assumptions from $LTL_f$ and considers events as pointwise and non-overlapping activities. *Secondly*, while the former only supports conditions on the activity labels, $xtLTL_f$ also supports predicating on the conditions for the payload values (expressed as key-value pairs) associated with the specific events [28], as well as supporting binary predicates to be tested across activated and targeted conditions similarly to $\theta$-joins. Recent extensions of Allen's algebra aimed at supporting single data conditions over single events [40]. *Thirdly*, such algebra only expresses temporal correlations between two single events, albeit expressed with a duration and a termination time, and can predicate natively neither the eventuality nor the necessity of some properties to occur in a trace (e.g., globally and future) from a given instant in time.

Concerning the temporal relational algebra [22] defined over temporal relational databases [47] (also referred to as *temporal modules* [21]), it mainly proposes timestamp transformation operations currently supported by Oracle Cloud [48] as well as windowing functions, thus retaining the entities and relationships occurring within a window time frame. This allows the slicing of a temporal module into a finite sequence of finite database states, where such a snapshot sequence can be ascribed to a single trace and each event can be mapped to a single database state [49]. Despite time being considered as a first citizen within these operators, no operator of such an algebra temporally correlates entities at different timestamps while also requiring the eventuality or the necessity for a specific condition within a given lapse of time. An orthogonal contemporary approach attempted at mapping $LTL_f$ to TSQL2 [50], a de facto extension of SQL for querying temporal modules [51]. Differently from the approach mentioned above, this preserved $LTL_f$ temporal operators such as Until ($\mathcal{U}$); as the authors preceded the definition of $LTL_f$ extensions considering data payload conditions [3,28], these are not considered in their transformation. Furthermore, as these temporal modules represent one single distinct trace as a result of temporal snapshotting of a single database into multiple distinct states, they cannot be effectively used to run a single formal verification task over numerous traces as per our proposed approach, as this would require running a single TSQL2 query over multiple databases, one for each log trace. In fact, our solution can assess multiple traces simultaneously by leveraging an extended relational representation to the one initially described in [17].

## 3. Proposed Derived Operators

*Similarly to the definition of the derived operators in relational algebra, we now provide the definition of our proposed operators extending $xtLTL_f$ by expressing those in terms of the ones already known in such a temporal algebra. These are then defined in Equations* (3), (5), (7) *and* (9).

### 3.1. AndAltFuture

We want this operator to seek all the instants of time when an event activates the Declare clause while the target follows anytime in the future, while requiring that no further activation occurs between these two events. This operator aims to optimize the AltResponse(A,B) clause and can be then expressed in terms of basic $\text{xtLTL}_f$ operators as follows:

$$\text{AndAltFuture}^{\tau}_{\Theta}(\rho, \rho') \overset{\text{def}}{=\!=} \text{And}^{\tau}_{\Theta}\left(\rho, \text{Next}\left(\text{Until}^{\tau}_{\textbf{True}}\left(\text{Not}^{\tau}(\rho), \rho'\right)\right)\right) \tag{3}$$

By implementing this operator from scratch, we want to avoid running the costly computation of the timed $\text{Until}^{\tau}$ unless the activation condition associated with the intermediate result returned as $\rho$ is satisfied. Furthermore, we want to avoid explicitly computing the negation of the activation condition and express this by explicitly checking that, given any activating event in $\sigma^i_j$ in $\rho$ with an immediately following targeting one $\sigma^i_k$ in $\rho'$ with $|\sigma^i| > k > j$, no other events $\sigma^i_{j+h}$ in $\rho$ with $j + h < k$ shall occur. We can then express the aforementioned Declare clause in terms of the recently defined operator as follows:

$$\text{Globally}^{\tau}\left(\text{Or}^{\tau}_{\textbf{True}}\left(\text{Not}^{\tau}(\rho), \text{AndAltFuture}^{\tau}_{\textbf{True}}(\rho, \rho')\right)\right) \tag{4}$$

where $\rho = \text{Activity}^{\mathbb{S}, \tau}_A(\text{A})$ and $\rho' = \text{Activity}^{\mathbb{S}, \tau}_T(\text{B})$ under the dataless assumption.

**Example 2.** *With reference to the log in Equation (1), AltResponse(`redirect`, `examine`) requires that a patient redirected to a given department shall be examined before being further redirected. This constraint satisfies all the traces within that equation. By considering only the events from the second trace, in our previous `xtLTL`$_f$ solution we have intermediate results $\rho = Activity_A(\texttt{redirect}) = \{\langle 1, 1, [A(1)]\rangle\}$ for the activation condition and $\rho' = Activity_T(\texttt{examine}) = \{\langle 1, 3, [T(3)]\rangle\}$ for the target one. The timed Until $\rho'' = Until^{\tau}_{True}(\neg\rho, \rho')$ returns:*

$$\{\langle 1, 0, [T(1)]\rangle, \ldots, \langle 1, 3, [T(3)]\rangle, \langle 1, 4, []\rangle\}$$

*as each event in `xtLTL`$_f$ can only witness a future event, and $\rho''' = Next^{\tau}(\rho'')$ returns:*

$$\{\langle 1, 1, [T(3)]\rangle, \ldots, \langle 1, 2, [T(3)]\rangle, \langle 1, 3, []\rangle\}$$

*Hence, $And^{\tau}_{True}(\rho, \rho''')$ returns just $f = \{\langle 1, 1, [M(1,3)]\rangle\}$, while witnessing that, from that time onwards, both activation $A(1)$ and target $T(3)$ condition will occur from the same event 1. The rest of the events will be returned via $\neg\rho$, which are finally grouped-by temporally via untimed Globally. Before running it, we previously ran the timed Until operator independently from the occurrence of $\rho''$ in a trace.*

*On the other hand, our new AndAltFuture operator directly returns $f$ after taking as an argument $\rho$ and $\rho'$; this scans the events in $\rho'$ occurring after each occurrence of events in $\rho$ while immediately discarding the events in $\rho$ containing another redirect event in between. This reduces the memory footprint and the number of scans from our previous query plan.*

### 3.2. AndAltWFuture

Reflecting upon the definition of AltPrecedence(A,B) which this operator is aiming to optimize, we can observe that implementing an ad hoc operator AndAltWFuture for this might provide even greater optimization, as we might as well avoid checking the global absence of A-labelled events if no B occurs in a trace after an A. Therefore, this operator acts as an extension of the former by either requiring an alternate occurrence between activation

and target condition, as previously, or requiring the absence of any future activation if no targeting event is expected to occur. $\mathsf{AndAltWFuture}_\Theta^\tau(\rho, \rho')$ can be then defined as follows:

$$\mathsf{And}_\Theta^\tau\left(\rho, \mathsf{Next}\left(\mathsf{Or}_{\mathbf{True}}^\tau\left(\mathsf{Until}_{\mathbf{True}}^\tau(\mathsf{Not}^\tau(\rho), \rho'), \mathsf{Globally}^\tau(\mathsf{Not}^\tau(\rho))\right)\right)\right) \tag{5}$$

We can now express AltPrecedence(A,B) by replacing, in the original $\mathtt{xtLTL_f}$ Declare semantics, the previous equation with the currently introduced operator, thus obtaining:

$$\mathsf{Or}_{\mathbf{True}}^\tau\left(\mathsf{Until}^\tau\left(\mathsf{Not}^\tau(\rho'), \rho\right), \mathsf{Globally}^\tau\left(\mathsf{Or}_{\mathbf{True}}^\tau\left(\mathsf{Not}^\tau(\rho), \mathsf{AndAltWFuture}_{\mathbf{True}}^\tau(\rho, \rho')\right)\right)\right) \tag{6}$$

*3.3. AndNext*

This operator aims to optimize the ChainResponse operator by reducing the data access by accessing the `ActivityTable` just for the activation condition. This makes this operator intrinsically unary, as the target condition, both in terms of data predicate and activity label, has to be provided as additional arguments for the operator alongside the $\Theta$ correlation condition for dataful scenarios. To check whether the target condition occurs immediately after the operand's current event, we need to check whether it is associated with an activity table and whether it satisfies a predicate $q$. This can be then expressed in $\mathtt{xtLTL_f}$ in terms of the following derived operator:

$$\mathsf{AndNext}_{B,q,\Theta}^\tau(\rho) \overset{\text{def}}{=\!=} \mathsf{And}_\Theta^\tau\left(\rho, \mathsf{Next}^\tau(\mathsf{Atom}_T^{\mathfrak{S},\tau}(B, q))\right) \tag{7}$$

At this stage, we can then express the semantics associated to the Declare template ChainResponse(A,B) as follows:

$$\mathsf{Globally}^\tau\left(\mathsf{Or}_{\mathbf{True}}^\tau\left(\mathsf{Not}^\tau(\rho), \mathsf{AndNext}_{B,\mathbf{True},\mathbf{True}}^{\mathfrak{S},\tau}(\rho)\right)\right) \tag{8}$$

where $\rho = \mathsf{Activity}_A^{\mathfrak{S},\tau}(A)$ in a dataless scenario.

*3.4. NextAnd*

The second operator aims at optimizing ChainPrecedence(A,B) similarly to the previous one, but with a swapped temporal occurrence. Please observe that negating the fact that an event shall occur after another can be expressed in terms of all the events occurring at the end of a trace and all of the events not matching the activation condition a when occurring in a non-first position. So, ChainPrecedence is usually represented as:

$$\mathsf{Globally}^\tau\left(\mathsf{Or}_{\mathbf{True}}^\tau\left(\mathsf{Or}_{\mathbf{True}}^\tau\left(\mathsf{Last}^{\mathfrak{S},\tau}, \mathsf{Next}^\tau(\mathsf{Not}^\tau(\rho))\right), \mathsf{And}_{\mathbf{True}}^\tau\left(\mathsf{Next}^\tau(\rho), \rho'\right)\right)\right)$$

where $\rho = \mathsf{Activity}_A^{\mathfrak{S},\tau}(A)$ and $\rho' = \mathsf{Activity}_T^{\mathfrak{S},\tau}(B)$ in a dataless scenario. After compactly representing the subexpression in the second row of the previous definition, as follows:

$$\mathsf{NextAnd}_{B,q,\Theta}^\tau(\rho) \overset{\text{def}}{=\!=} \mathsf{And}_\Theta^\tau\left(\mathsf{Next}^\tau(\rho), \mathsf{Atom}_T^{\mathfrak{S},\tau}(B, q)\right) \tag{9}$$

we aim to optimize this last declarative clause by using this last introduced operator by rewriting the semantics associated to `ChainPrecedence(A,B)` as such:

$$\mathsf{Globally}^\tau\left(\mathsf{Or}_{\mathbf{True}}^\tau\left(\mathsf{Or}_{\mathbf{True}}^\tau\left(\mathsf{Last}^{\mathfrak{S},\tau}, \mathsf{Next}^\tau(\mathsf{Not}^\tau(\rho))\right), \mathsf{NextAnd}_{B,\mathbf{True},\mathbf{True}}^\tau(\rho)\right)\right) \tag{10}$$

Please observe that the intended optimization induced by these operators can be considered as non-trivial, as these do not directly subsume the entire xtLTL$_f$ semantics associated to a template, rather than optimizing a specific part.

## 4. Algorithmic Implementation

*We discuss the implementation of the previously introduced operators outlined in Algorithm 1, thus justifying their definition as novel derived operators. For each of them, we briefly discuss their computational complexity and compare it to the expected theoretical speed-up not considering the cost of memory allocation and page-faults.*

---

**Algorithm 1** Newly proposed xtLTL$_f$ operators.

---

1: **function** ANDALTFUTURE$_\Theta^\tau(\rho, \rho')$
2:    **for all** $\langle i, j, L \rangle , \langle i, k, L' \rangle \in (\rho \times \rho')$ **s.t.** $j < k$ **do**
3:        **if** $\nexists h > 0. \langle i, j + h, L \rangle \in \rho$ **s.t.** $j + h < k$ **then**
4:            **if** $L' \neq \varnothing$ **and** $L \neq \varnothing$ **and** $\Theta \neq$ **True then**
5:                $L'' \leftarrow \{M(j', k') | \Theta(\sigma_{j'}^i, \sigma_{k'}^i), A(j') \in L, T(k') \in L'\}$
6:                **if** $L'' \neq \varnothing$ **then yield** $\langle i, j, L'' \rangle$
7:            **else yield** $\langle i, j, L \cup L'' \rangle$
8:            **end if**
9:        **end if**
10:    **end for**

11: **function** ANDALTWFUTURE$_\Theta^\tau(\rho, \rho')$
12:    **for all** $\langle i, j, L \rangle \in \rho$ **do**
13:        **for all** $\langle i, k, L' \rangle \in \rho'$ **s.t.** $j \leq k$ **do**
14:            **if** $\nexists h > 0. \langle i, j + h, L \rangle \in \rho$ **s.t.** $j + h < k$ **then**
15:                **if** $j = |\sigma^i| - 1$ **continue;**
16:                **if** $L \neq \varnothing$ **and** $L' \neq \varnothing$ **and** $\Theta \neq$ **True then**
17:                    $L'' \leftarrow \{M(j', k') | \Theta(\sigma_{j'}^i, \sigma_{k'}^i), A(j') \in L, T(k') \in L'\}$
18:                    **if** $L'' \neq \varnothing$ **then yield** $\langle i, j, L'' \rangle$
19:                **else yield** $\langle i, j, L \cup L'' \rangle$
20:                **end if**
21:            **end if**
22:        **end for**
23:        **if** $\nexists k, h. \langle i, k, L' \rangle \in \rho' \wedge \langle i, h, L'' \rangle \in \rho \wedge j < k, j < h$ **then**
24:            **yield** $\langle i, j, L \rangle$
25:        **end if**
26:    **end for**

27: **function** ANDNEXT$_{B,q,\Theta}^\tau(\rho)$
28:    **if** $\nexists \sigma^i \in \mathfrak{S}, \sigma_j^i \in \sigma^i.\lambda(\sigma_j^i) = B$ **then return** $\varnothing$
29:    **for all** $\langle i, j, L \rangle \in \rho$ **s.t.** $j < |\sigma^i| - 1$ **and** $\lambda(\sigma_{j+1}^i) = B$ **do**
30:        $L' \leftarrow L \cup \{T(j+1)\}$
31:        **if** $\Theta \neq$ **True then**
32:            **if** $L \neq \varnothing$ **and** $\nexists A(k) \in L.\theta(\sigma_k^i, \sigma_{j+1}^i)$ **then continue**
33:            **else** $L' \leftarrow \{M(k, j+1) | A(k) \in L\}$
34:        **end if**
35:        **if** $q \neq$ **True** $\vee q(\sigma_{j+1}^i)$ **then yield** $\langle i, j, L' \rangle$
36:    **end for**

37: **function** NEXTAND$_{B,q,\Theta}^\tau(\rho)$
38:    **for all** $\langle i, j + 1, L \rangle \in \rho$ **s.t.** $j \geq 0$ **and** $\lambda(\sigma_j^i) = B$ **do**
39:        $L' \leftarrow L \cup \{T(j)\}$
40:        **if** $\Theta \neq$ **True then**
41:            **if** $L \neq \varnothing$ **and** $\nexists A(k) \in L.\theta(\sigma_k^i, \sigma_j^i)$ **then continue**
42:            **else** $L' \leftarrow \{M(k, j) | A(k) \in L\}$
43:        **end if**
44:        **if** $q \neq$ **True** $\vee q(\sigma_j^i)$ **then yield** $\langle i, j + 1, L' \rangle$
45:    **end for**

---

### 4.1. AndAltFuture

As all the intermediate results in the KnoBAB pipeline are always sorted by ascending trace and event id, we can scan all the events within the same trace where the targets follow the activations in linear time similarly to the timed *and* operator, despite this being expressed in pseudocode with a cross product for simplifying the overall notation (Line 2). We then consider all the events in the same trace having no immediate subsequent event in $\rho$ prior to the occurrence of the next event in $\rho'$; this can be simply checked in $\rho$ by determining that

the next record appearing in $\rho$ after $\langle i, j, L \rangle$ has an event id less than $k$ (Line [3]). If there is a non-trivially true $\Theta$ predicate, we also impose that at least one activation occurring after or at $\sigma_j^i$ and at least one target occurring after or at $\sigma_k^i$ matches with $\Theta$ (Line [6]). Otherwise, we compute no match, and we straightforwardly collect the activation and target conditions from both events (Line [7]). In the code, we explicitly injected an early-stopping condition avoiding testing subsequent events in $\rho'$ within the same trace as soon as we detect one event in $\rho$, invalidating the condition at Line [3]. By considering the time complexities for each $\mathtt{xtLTL_f}$ operator in Section [2.3], we can argue that the time complexity associated with computing this operator as in the previous section without the aforementioned computation is totalled to $O(|\rho| + (||\mathfrak{S}|| - |\rho|)^2 |\rho'| + 2((||\mathfrak{S}|| - |\rho|) + |\rho'|))$, where $||\mathfrak{S}|| = |\mathfrak{S}|\epsilon$. On the other hand, by assuming to always scan each trace quadratically of length $\epsilon$ for each event in $\rho$, we obtain the time complexity of $O(|\rho|\epsilon^2/2 + |\rho'|)$ for the derived operator when implemented as per the previous discussion. If we assume that $\rho$ and $\rho'$ are associated with a single activity label, as per the scenario in Declare, where the number of events and the activity labels are uniformly distributed such that $\#\mathtt{a} \approx |\mathfrak{S}|\epsilon/|\Sigma|$ for each $\mathtt{a} \in \Sigma$, we can derive that the provided algorithm always provides a positive speed-up if compared to the original formulation in Equation (3).

### 4.2. AndAltWFuture

This algorithm works similarly to the previous, where we relax the until condition with a weak until, thus also admitting an absence of activation conditions after the first occurrence (of an activation) if no further target events are present (Line [23]). Even in this scenario, we have a similar time complexity to the previous, while the original formulation in Equation (5) introduced an additional overhead to the previous by computing an additional timed disjunction and the global computation over the negation of the possibly activating events. Therefore, we expect an even greater speed up for this latest operator.

### 4.3. AndNext

As previously observed in the formal definition of this operator, we transformed this into an unary operator where, instead of retrieving two sets of events associated with two activity labels, we just scan one of the two. Before starting any form of scan, we immediately return if, after a $O(|\mathfrak{S}|)$ scan of the CountTable, we detect that no event is associated with the target condition (Line [28]). Otherwise, we consider only events both coming from traces containing an event with activity label $\mathtt{B}$ and not being at the end of the trace, and for which the immediately next event is associated to an activity label $\mathtt{B}$ as a target condition ($T(j+1)$, Line [29]); we implementationally further enhanced this by completely skipping any test whether the event resides in a trace where no $\mathtt{B}$ event resides. If $\Theta \neq \mathbf{True}$, then we also have to guarantee that each activation condition appearing in $\rho$ should match with the target event at time $j + 1$ (Line [33]) and, upon provision of $q$, the target condition should also match with this (Line [35]). The computational complexity of this operator is in $O(|\rho| + |\mathfrak{S}|)$ and, if we are taking into account the accessing time to the immediately following event, if any, we obtain a time in $2|\rho| + |\mathfrak{S}|$. If compared to the time complexity of Equation (7) of $|\rho| + 2|\rho'|$, we then obtain a positive speed up, i.e., $\frac{|\rho| + 2|\rho'|}{2|\rho| + |\mathfrak{S}|} > 1$, for $|\rho'| > |\rho|/2$ and $0 < |\mathfrak{S}| < 2|\rho'| + |\rho|$.

### 4.4. NextAnd

This other operator works similarly to the previous, where we are checking instead the immediately preceding event instead of looking at the immediately following one, thus requiring that each element of interest in $\rho$ shall never be at the beginning of the trace. The same considerations over speed-up and time complexity follow from the previous algorithm.

After associating each of the novel operators in the aforementioned algorithmic implementation, Equations (4), (6), (8) and (10) will then provide the semantics generating

the query plan as `Proposed` in this current paper, while the direct translation of the $LTL_f$ expressions in Table 1 to the operators outlined in Section 2.2.2 provides the `Original` formulation of the query plan also in [28], where none of the previous algorithms are used.

## 5. Empirical Evaluation

*Given that the aim of our derived operators is to enhance formal verification tasks conducted over temporal clauses expressed in Declare, we compare the different running times of carrying out formal verification tasks over our previous set of operators as well as by replacing those with our currently proposed derived ones, while focusing on benchmarking formal verification tasks over specifications written in Declare. We discard from our evaluation the benchmark of the single operator, as this is insufficient to remark on their adequacy in enhancing formal verification tasks in Declare. Thus, we compare different query plans being generated from different Declare semantics being specified at runtime through the `queryplan "name" {...}` query. With this, achieving a positive speed-up in Declare formal verification tasks as in our previous work [28] by using the proposed operators will tell us that, under specific data conditions, the original `xtLTLf` query plan associated with the declarative clauses available in KnoBAB constitutes the major computational bottleneck. Having a negligible speed-up will likely remark other components in the query plan dominating the overall running time, while having a negative speed-up only on specific data conditions will motivate some future work on hybrid algorithms, thus allowing us to choose between different algorithms for specific temporal operators depending on the data distribution within the loaded dataset [52].*

Our benchmarks exploited a Dell Mobile Precision Workstation 5760 on Ubuntu 22.04: Intel® Xeon(R) W-11955M CPU @ 2.60 GHz × 16, 64 GB DDR4 3200 MHz RAM. We took two real-world datasets and a synthetic one for our experiments, both being dataless. The first real dataset (Hospital) monitors the patient flow and different medical procedures to which the patients in question were subjected; each trace tracks a single patient from his hospitalization to dismissal, and each activity label describes the name associated to such phases [10]. The second one (Cybersecurity) provides the auditing step of different malware, where each trace represents a single malware being audited, while each activity label identifies one single system call event being audited as invoked by the malware [6,7]. The synthetic dataset was derived from temporal graphs generated by FoodBroker [11] while describing trades and shipments of goods mediated by a brokerage company. For each `GraphTransaction`, we sort all the vertices describing an event occurring at a specific `date`, thus also including `creation` timestamps. For vertices describing a ticket being filed by a client raising a complaint, we return an activity label associated with the type of complaint (`problem`); otherwise, we keep the original vertex label. We then collect the set of temporally ordered activity labels and represent those as log traces. The updated FoodBroker codebase for generating event logs is also available online (https://github.com/jackbergus/foodbroker/, accessed on 3 January 2024).

For each dataset, we then obtain the sampled trace length distribution, and we sample sub-logs of various sizes while trying to abide by the trace distribution from the original dataset, notwithstanding their skewness. For the first and third (or second) datasets, we sample the logs so that their sizes are powers of ten (or nine) while always guaranteeing that each sub-log $|\mathfrak{S}_h| = 10^h$ (or $|\mathfrak{S}_h| = 9^h$) is always a subset of any larger sub-log. We also keep the original log as the last sample dataset. This random sampling mechanism is required to better assess the scalability of the proposed operator's implementation while guaranteeing an approximation of the original trace length distribution across the board to guarantee similar running time conditions. Figure 2 reports the sample PDF trace length for each of the sampled logs alongside the size of each sample. The FoodBroker synthetic dataset contains the shorter traces (Figure 2a); all the sampled logs except the first one have a maximum trace length of 24, while the first sublog has a maximum trace length of 21. On the other hand, the first two smaller log samples of the real-world Hospital dataset (Figure 2b) have traces with a maximum length of 1200, while the remaining two have a maximum trace length of 1814. The Cybersecurity dataset (Figure 2c) contains the

longest traces, having a maximum trace length of $1.23 \times 10^6$ for the smaller two sub-logs and of $1.76 \times 10^6$ for the remaining ones. This information will soon become relevant while conducting our following analysis of the algorithmic speed-ups given by our proposed derived operators while performing formal verification over the models described in the following paragraph.

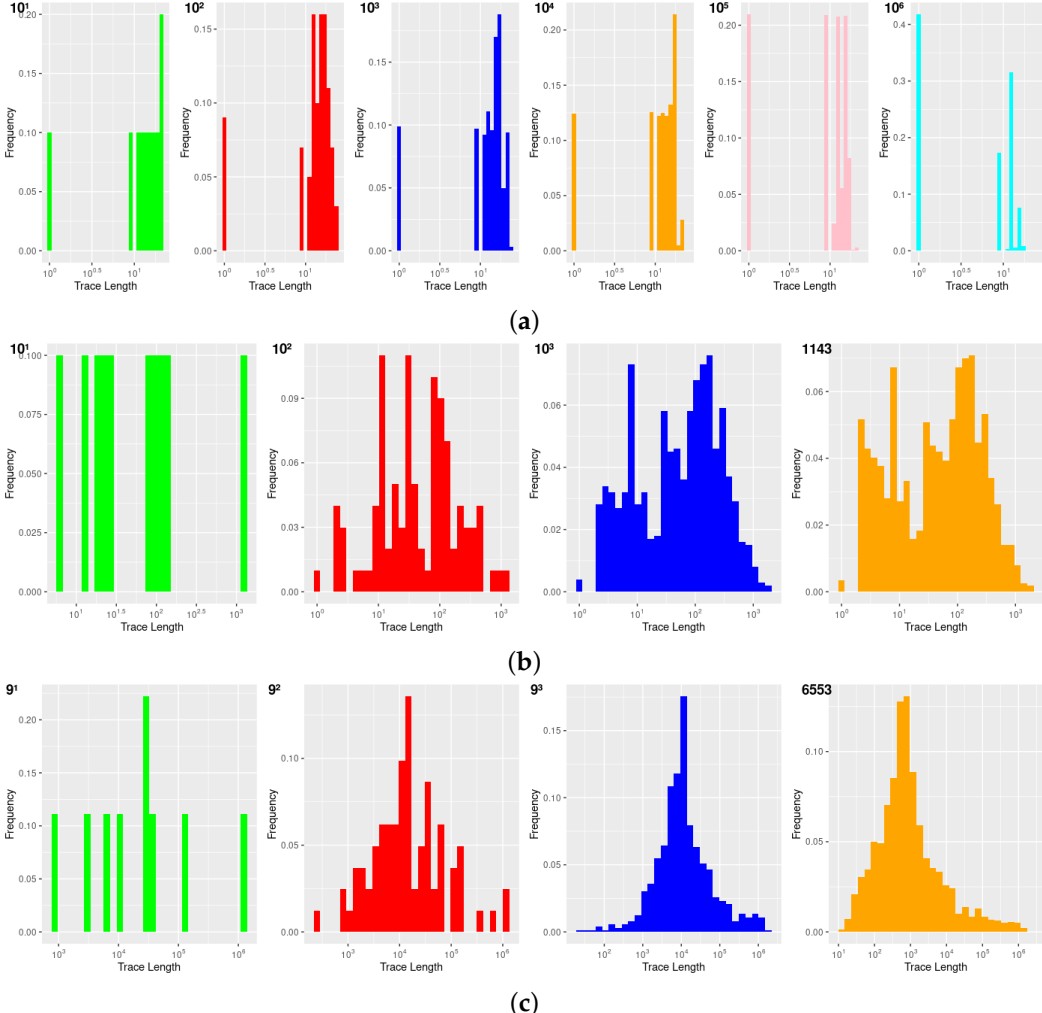

**Figure 2.** Sampled probability density function associated with the length of the traces for each sub-log extracted from each original dataset: (**a**) FoodBroker, (**b**) Hospital, and (**c**) Cybersecurity.

Given that we aim to test these newly introduced $\text{xtLTL}_f$ operators in the context of a Declare-based formal verification task when $\text{xtLTL}_f$ is used to represent its semantics, we generate four specifications $\Phi_1^c, \ldots, \Phi_4^c$ for each declarative clause of interest $c$, AltPrecedence, AltResponse, ChainPrecedence, and ChainResponse, where each $\Phi_i^c$ contains exactly $i$ binary clauses determined by instantiating an activity label among the most frequently occurring ones within the smaller sub-log. We then use the same specifications generated for the smaller log and the greater sub logs, thus comparing the running times for each sub-log over the same Declare specifications. We then use the specifications to conduct a formal verification task via a `model-check...` query. The resulting logs and specifications are freely available online (https://osf.io/6y8cv/, accessed on 3 January 2024).

Last, as our previous work already showed that computing such queries on top of relational databases such as PostgreSQL with shorter traces leads to a greater running time than running similar queries over KnoBAB, we just focus on comparing the results from our previous implementation with the ones after applying the changes discussed in this paper.

With reference to Figure 3, AndAlt∗ operators can be deemed responsible for evidently outperforming the proposed query plan if compared to the one from the previous implementation, as they lead to an associated speed-up always strictly greater than one. Our previous definition of the Declare operators is greatly affected by the number of clauses within the model, which becomes even more apparent when the maximum and average trace length $\epsilon$ per sampled log increases. On the other hand, running our former formal verification query plan for AndAlt∗ clauses over the Cybersecurity dataset always took more than one 1H ($3.6 \times 10^6$ ms), thus demonstrating an increased running time for the original query plan strategy when longer traces occur. We stopped recording the running time, as the overhead introduced by the intermediate operators for carrying out the actual matching between activation and target conditions was strikingly evident, while our proposed operators could instead carry out the formal verification task within one minute. Although no out-of-memory exceptions were observed before the timeout, these were clearly observed in larger specifications and log sizes, thus clearly demonstrating KnoBAB's limits as a main memory engine by not maintaining the query intermediate results in secondary memory. Despite the code allowing the clearing of intermediate caches to be run to free extra memory, this only partially addresses the out-of-memory failure for larger specifications. Overall, this demonstrates that this proposed extension for AndAlt∗ operators outperforms our previous query plan definition, as also expected from our previous analysis concerning the overall theoretical time complexity.

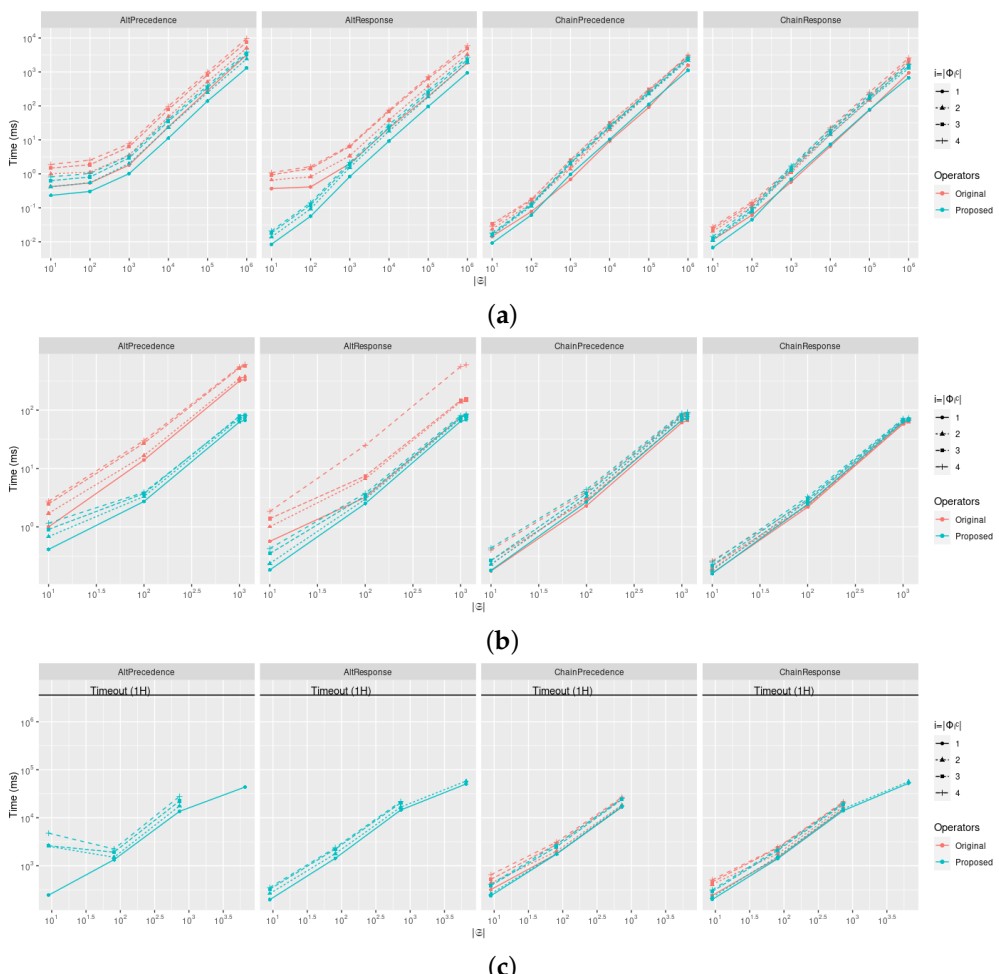

**Figure 3.** *Cont.*

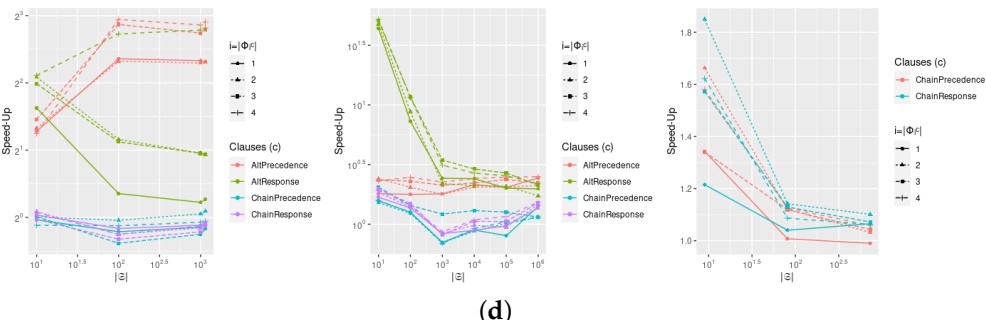

**(d)**

**Figure 3.** Comparing the proposed implementation of the derived operators with the previous implementation given in KnoBAB. (**a**) FoodBroker dataset; (**b**) Hospital dataset; (**c**) Cybersecurity dataset. (**d**) Datasets' speed-up: (**left**) FoodBroker, (**center**) Hospital, and (**right**) Cybersecurity.

Chain∗ operators provide a more convoluted scenario to be examined carefully. First, we observe a clear trend correlating datasets with longer traces with an overall increase in speed-up. In fact, the Hospital datasets exhibit more speed-ups compared to the FoodBroker one, where the recently proposed operators yield comparable or underperforming running times. Notwithstanding the former, we can clearly observe that the recently proposed operators consistently outperform our previous solution over the Cybersecurity dataset. Differently from our previous set-up, we can now observe that the original formulation of the declarative clauses without the currently presented operators now runs out of memory before hitting the 1H timeout for the larger sample, being the full dataset, while our solution still manages to carry out some temporal formal verification tasks over specifications containing fewer clauses. Last, we consistently observe that such operators still provide greater speed-ups over datasets with smaller log sizes, thus providing theoretical validation to our speed-up equations for such operators. This postulates the need for such operators while dealing with massive datasets, while advocating the usage of hybrid algorithms for switching between the previous solution and the currently proposed one.

## 6. Conclusions and Future Works

This paper proposes an extension to our previous work on KnoBAB by optimizing our previously proposed query plan by introducing novel algebraic temporal operators expressing formal verification tasks on column database storages in main memory. As a consequence, we extended our temporal algebra $\mathrm{xtLTL_f}$ with four novel operators, subsuming entire $\mathrm{xtLTL_f}$ expressions which before could only be represented in terms of combinations of costly basic operators. Preliminary results over such operators provide non-negligible speed-up to the formal verification tasks over realistic datasets, where several events are audited and collected in a larger collection of traces.

Despite these experiments demonstrating the efficiency of carrying out formal verification computations on columnar databases implemented as a main memory engine, the consistent out-of-memory faults that we experienced over larger collections of data containing more events (i.e., longer traces) encourage us to store the intermediate query results in secondary memory, as customary for off-the-shelf databases such as PostgreSQL. We see this as the last required step for fully supporting real data alongside the orthogonal operator optimization, as discussed in the present paper. Despite the fact that putting this solution in place will come at the detriment of overall performance, this will guarantee the carriage of the entire formal verification computation. This drawback might be alleviated by determining at runtime whether to represent intermediate results in primary or secondary memory depending on the log and trace size. Another possible way to alleviate such a problem is to re-implement the overall pipeline using a pull-based strategy [33] when operations are not run concurrently. Another way to challenge this primary memory limitation would be migrating the proposed architecture over Oracle Cloud [48], which already supports traditional time-transactional database operations compatible with the aforementioned *temporal modules*. While doing so, we will be walking in the footsteps

of previous literature [50] by attempting to rewrite dataful $xtLTL_f$ specifications into the supported temporal extension of SQL.

The current experiment noted the optimality of the proposed operators when dealing with datasets with longer traces (i.e., greater $\epsilon$). Future work will consider the possibility of defining hybrid algorithms [27] over the operators, optimizing Chain∗ clauses by empirically determining the table size threshold over which we prefer the derived operators over the original. As an orthogonal approach, we will also define the "dual" operators for ANDNEXT and NEXTAND so as to start scanning from the target condition while moving backwards towards any existing activation condition when the number of targets is deemed to be fewer than the activations. Our future works will also aim to further benchmark these operators in the context of *dataful* logs, where events are also associated with a payload expressed as a key-value pair as in customary semi-structured data formats. These works will then outline the overhead required to compute a $\Theta$ correlation condition between activation and target event.

Previous research on temporal modules demonstrated the possibility of expressing $LTL_f$ specifications when traces have multiple events occurring at one specific point in time [50]; the current theoretical literature on conformance checking suggests that this is actually possible by representing each single event as a conjunction of multiple mutually exclusive events, thus obtaining the characterization of composite events. However, realizing this in practice for events with distinct labels would require a drastic overhaul of KnoBAB's relational representation, as the current architecture is focused on the linear representation of each individual trace. Future work will therefore contemplate the possibility of extending the current relational model with an object-oriented one [53], better supporting the nesting and composition of objects, a feature also required for coalescing multiple events in a single instant in time.

Finally, an interesting outcome of these observations on relational databases would be the application of such an algebra in the context of temporal graphs [54], thus enabling the efficient temporal verification under this different data representation. Despite the recent attempt at representing logs as temporal graphs [55], the aforementioned is still a desideratum, as no graph temporal operator for expressing formal verification tasks is currently known. Differently from the previously pursued approach [56], this will then require us to define tailored temporal operators for graph query languages similarly to $xtLTL_f$.

**Funding:** This research received no external funding.

**Institutional Review Board Statement:** Not applicable.

**Informed Consent Statement:** Not applicable.

**Data Availability Statement:** The dataset is available at https://osf.io/6y8cv/, accessed on 3 January 2024.

**Conflicts of Interest:** The author declares no conflict of interest.

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
