# Peer review of "Streamlining Temporal Formal Verification over Columnar Databases"

_information, doi:10.3390/info15010034_

Round 1

Reviewer 1 Report

Comments and Suggestions for Authors

Table 1: formal notation.

However, the applied notations are fully explained in the text later on in the paper, it would be nice to refer to the exact places of the explanation in the Table or footnote to the table ( the "W", 'Weak until', and "U" , 'Until'notation). Because even the referred "DECLARE" article does not contain all the notations.

Two concepts should be explained in detail or through definitions, namely, 'dataful' and 'dataless'. These concepts cannot be found in the cited articles.

The argumentation of time complexity is hard to follow. It would be better to map the proposed search algorithm onto the fundamental searching algorithm (linear, etc.). Then the fundamental searching algorithm and proposed approaches can be compared easily, and the estimation could be checked.

|ρ| notation should be defined accurately, it can be guessed that it is the length of a sequence, but it is nowhere specified explicitly

The genre of the research should be made more explicit. The research focuses on a performance analysis that is clear from Sections 4. and 5.

The model checking formulas conceptualized through linear temporal logic is the subject of the performance analysis. The formulation of model checking seems like a “routine” model checking task, and it does not contain proper novelty.  However, these formulations are an important pre-condition for further analysis. A figure about the architecture can help understand the structure of the system.

Comments on the Quality of English Language

There is no serious grammatical or spelling error.

Author Response

We thank the reviewer for the constructive comments that we hope helped to shape the paper better. Please find our answers in the attachment.

Reviewer 2 Report

Comments and Suggestions for Authors

The abstract summarizes the content of the article and lists the achieved results and benefits of the work.

The review of the literature and the current state of knowledge is comprehensive enough.

The symbols used are clearly defined.

Conclusions and Future Works are also sufficient.

From this point of view, I have nothing to criticize the author or recommend for improvement.

On the technical side:

1. I recommend correcting the parentheses in the formulas - or use their different sizes. Different sizes of brackets will improve orientation in the text for readers. Do at least like in formula 6.

2. Delete the Section title „ 7. Patents“

3. Captions under the images are not legible - fix them

An interesting article that deserves to be published.

Author Response

(The authors gave the same response as above.)

Reviewer 3 Report

Comments and Suggestions for Authors

This paper is really interesting and provides valuable new strategies for managing and evaluating temporal data. Relational databases still provide sufficient power, and temporal extensions are now an inseparable part of the technology. 

It would be worth to deeply reference temporal database approaches, because a significant progress was been done in the last year: https://ieeexplore.ieee.org/search/searchresult.jsp?queryText=temporal%20databases&highlight=true&returnFacets=ALL&returnType=SEARCH&matchPubs=true&sortType=newest

Or even books are relevant for the reference:

https://www.amazon.com/Developing-Robust-Oriented-Applications-Oracle-ebook/dp/B0BZJG8G7Q/ref=sr_1_1?crid=3BLBAKQN2NBE5&keywords=michal+kvet&qid=1681138050&sprefix=michal+kve,aps,219&sr=8-1&language=en_US&currency=EUR 

https://www.amazon.com/Using-Temporal-Datalog-Databases-Evolving/dp/1019420022/ref=sr_1_1?crid=KYUHPIASFZT&keywords=temporal+databases&qid=1704198146&sprefix=temporal+databases%2Caps%2C176&sr=8-1

These new references would provide a deeper context of the temporal data processing. 

Besides, I recommend adding a section discussing the scalability of the solution, how it would work for a significant large data set? The experiments and results should be more properly discussed. At least provided figures and tables should be discussed. 

Please state also the limitation of the research and future research perspectives. 

Text in fig. 2 is cut off. 

Reconsider the references to be more up-to-date. 

Comments on the Quality of English Language

Text in fig. 2 is cut off. 

Author Response

(The authors gave the same response as above.)
